# Comprehensive Overview of Ketone Bodies in Cancer Metabolism: Mechanisms and Application

**DOI:** 10.3390/biomedicines13010210

**Published:** 2025-01-16

**Authors:** Ziyuan Liang, Lixian Deng, Xiaoying Zhou, Zhe Zhang, Weilin Zhao

**Affiliations:** 1Key Laboratory of High-Incidence-Tumor Prevention & Treatment, Guangxi Medical University, Ministry of Education, Nanning 530021, China; zyliang0907@163.com (Z.L.); dlx233904288@163.com (L.D.); zhouxiaoying1982@foxmail.com (X.Z.); 2Life Science Institute, Guangxi Medical University, Nanning 530021, China; 3Department of Otolaryngology-Head and Neck Surgery, First Affiliated Hospital of Guangxi Medical University, Nanning 530021, China

**Keywords:** tumor, ketone bodies, metabolism, ketogenic diet, therapeutic approach

## Abstract

Reprogramming energy metabolism is pivotal to tumor development. Ketone bodies (KBs), which are generated during lipid metabolism, are fundamental bioactive molecules that can be modulated to satisfy the escalating metabolic needs of cancer cells. At present, a burgeoning body of research is concentrating on the metabolism of KBs within tumors, investigating their roles as signaling mediators, drivers of post-translational modifications, and regulators of inflammation and oxidative stress. The ketogenic diet (KD) may enhance the sensitivity of various cancers to standard therapies, such as chemotherapy and radiotherapy, by exploiting the reprogrammed metabolism of cancer cells and shifting the metabolic state from glucose reliance to KB utilization, rendering it a promising candidate for adjunct cancer therapy. Nonetheless, numerous questions remain regarding the expression of key metabolic genes across different tumors, the regulation of their activities, and the impact of individual KBs on various tumor types. Further investigation is imperative to resolve the conflicting data concerning KB synthesis and functionality within tumors. This review aims to encapsulate the intricate roles of KBs in cancer metabolism, elucidating a comprehensive grasp of their mechanisms and highlighting emerging clinical applications, thereby setting the stage for future investigations into their therapeutic potential.

## 1. Introduction

The reprogramming of energy metabolism is a hallmark of cancer, contributing to aggressive phenotypes and reducing therapeutic responses in patients [1]. Tumor cells exhibit distinct metabolic features compared to normal cells. They predominantly rely on glycolysis for energy production, even in the presence of oxygen (the Warburg effect), whereas normal cells mainly utilize oxidative phosphorylation. Additionally, tumor cells have an increased dependence on fatty acid synthesis and amino acid metabolism, particularly glutamine, to support rapid cell proliferation and metastasis [2]. Recently, ketone bodies (KBs) metabolism has gained considerable attention in cancer research. KBs are composed of acetoacetate (AcAc), acetone, and β-hydroxybutyrate (β-OHB), serving as a critical alternative fuel source to glucose during nutrient deprivation in diverse organisms, including eukaryotes and bacteria [3]. They are primarily synthesized in the liver mitochondria, with a little amount produced in astrocytes, cardiomyocytes, renal epithelial cells, and enterocytes [4]. In addition to acting as an energy source for extrahepatic tissues, KBs are a substructure for lipid and steroid biosynthesis in various tissues, such as the developing brain, lactating mammary gland, and liver [5]. They participate in regulating the synthesis of signaling molecules, post-translational protein modifications, inflammatory responses, and oxidative stress as well [6,7]. Increasing evidence indicates the prominent role of KB metabolism in various tumors [8]. A recent study has shed light on the mechanisms of metabolic reprogramming in cancer cells, highlighting the role of key signaling pathways—such as Reactive Oxygen Species (ROS), AMP-activated Protein Kinase (AMPK), Mitogen-activated Protein Kinase (MAPK), Phosphoinositide 3-kinase (PI3K), Hypoxia-inducible Factor 1α (HIF-1α), and Tumor Protein p53 (p53)—in regulating metabolic processes like glucose, lipid, and amino acid metabolism. These signaling pathways are critical for tumor progression, enabling cancer cells to adapt to the metabolic demands of rapid growth [9].

The ketogenic diet (KD), characterized as a high-fat, low-carbohydrate diet with adequate protein and calories, was originally developed in the 1920s for the treatment of intractable epileps [10]. The KD shifts metabolism by restricting carbohydrate intake, inducing ketosis, where fats become the primary energy source through ketone production. This diet prioritizes healthy fats—such as coconut oil, olive oil, butter, and nuts—along with moderate amounts of high-quality protein from sources like meats, fish, and eggs. Low-carbohydrate vegetables, including spinach, broccoli, and cauliflower, are also emphasized (Table 1). In contrast, high-sugar foods, starchy vegetables, and refined grains are typically avoided to maintain stable ketone levels. By reducing glucose availability and increasing KBs, KD forces tumor cells to shift their energy source. This metabolic switch drives tumor cells to rely on fatty acid oxidation, which is detrimental to their rapid proliferation [11]. Additionally, KD has been shown to improve the tumor microenvironment by reducing its acidity, thus inhibiting tumor cell invasiveness [12]. It also modulates immune function, suppressing tumor-associated immune suppressor cells (such as Tregs and M2 macrophages), which enhances anti-tumor immune responses [13]. To date, the efficacy of KD as a supportive therapeutic approach has been demonstrated in tumor treatment [14].

This review aims to summarize the latest findings regarding the importance of KBs and the major synthesis and catabolism enzymes involved in KB metabolism in different types of cancers, as well as their biological function in tumorigenesis. Additionally, we will explore future strategies for utilizing KD as an adjuvant therapy for tumors. To succinctly summarize the key points discussed in this review, the following table provides an overview of the dysregulation of KB metabolism enzymes in tumors, the potential roles of KBs in cancer, and the effects of KD as an adjunctive treatment for cancer. This table aims to offer a systematic summary to guide further research on the relationship between KB metabolism and cancer therapy (Table 2).

## 2. Dysregulation of KB Metabolism Enzymes in Tumor

KB synthesis begins in the cytoplasm of hepatocytes, where fatty acids are activated by coenzyme A (CoA) to form fatty acyl-CoA. With the help of carnitine palmitoyltransferase (CPT), fatty acyl-CoA is transported to the mitochondria of hepatocytes and oxidized by β-oxidation to acetyl-CoA. Acetoacetyl-CoA is formed when two acetyl-CoA molecules are condensed by acetoacetyl-CoA thiolase (ACAT1) [4]. The addition of a third acetyl-CoA molecule to AcAc-CoA, catalyzed by hydroxymethylglutaryl-CoA synthase (HMGCS2), results in the production of 3-hydroxymethylglutaryl-CoA (HMG-CoA). Subsequently, 3-hydroxy-3-methylglutaryl-CoA lyase (HMGCL) cleaves HMG-CoA to produce AcAc. While much of the AcAc pool is converted to the more stableβ-OHB by β-hydroxybutyrate dehydrogenase (BDH), the remainder is spontaneously decarboxylated to acetone. AcAc and β-OHB are then released into the circulation and taken up by extrahepatic cells for utilization via the monocarboxylic acid transporter (MCT). As hepatocytes cannot catabolize KBs, their utilization occurs in the mitochondria of extrahepatic tissues. In these tissues, β-OHB is converted back to AcAc by BDH. AcAc is then activated by 3-ketoacyl-coenzyme A transferase 1 (OXCT1), which adds a CoA molecule, and finally, ACAT1 catalyzes the cleavage of AcAc-CoA into two acetyl-CoA molecules. These acetyl-CoA molecules are then converted to citric acid by citrate synthase. Finally, citric acid enters the tricarboxylic acid (TCA) cycle and undergoes the electron transport chain reaction cascade to produce Adenosine Triphosphate (ATP). Additionally, ketolysis provides substrates for sterol synthesis, de novo lipogenesis, and hexosamine biosynthesis in various tissues [32]. Figure 1 displays the pathway of KB metabolism and the key enzymes involved.

To a certain extent, the dysregulation of KB metabolism is due to the differential expression of these enzymes. They could potentially serve as diagnostic and prognostic biomarkers of cancers, as well as therapeutic targets [3]. Tumors such as glioblastoma (particularly the glioblastoma multiforme subtype), triple-negative breast cancer, and colorectal cancer exhibit marked alterations in KB metabolism, especially under conditions of tumor hypoxia, glucose deprivation, or increased resistance to chemotherapy. In these contexts, KB metabolism serves as a critical alternative energy source [15,16,17].

### 2.1. ACAT1

ACAT1 is a mitochondrial enzyme involved in the metabolism of KBs, fatty acid β-oxidation, and isoleucine catabolism [33]. It facilitates the acetylation of Parkin, an E3 ubiquitin ligase, which leads to the suppression of cervical cancer through mitophagy [34]. ACAT1 also attenuates glycine decarboxylase activity, resulting in the inhibition of glioma growth [35]. Restoring ACAT1 expression increases endogenousβ-OHB levels and reverses epithelial–mesenchymal transition (EMT), indicating a potential tumor-suppressive function [36]. Conversely, elevated ACAT1 is believed to enhance the utilization of KBs as energy in breast cancer [37]. In various human cancer cells, including those from lung, leukemia, head and neck, and prostate cancers, ACAT1 activity is higher compared to corresponding normal cell lines [38]. In addition, ACAT1 stabilizes fatty acid synthase (FASN), which is essential for hepatocarcinogenesis [39]. It has been shown to promote tumourigenesis by inhibiting autophagy and scavenging ROS [40]. Consequently, ACAT1 inhibitors have been designed to inhibit tumor growth and metastasis [38]. These findings underscore the complex role of ACAT1 in cancer metabolism and its potential as a therapeutic target across multiple cancer types.

### 2.2. HMGCS2

The aberrant inactivation of HMGCS2 is significantly associated with tumor initiation and progression. For instance, a downregulation of HMGCS2 mediated by miR-107 was observed in hepatocellular carcinoma, accelerating tumor growth and migration. HMGCS2-knockout mice showed increased liver damage and inflammation after being fed a high-fat diet [41]. In addition, HMGCS2 decreased TNF-α-induced ROS production [42]. Chronic damage and inflammation are key pathogenic factors in hepatocellular carcinoma. Therefore, the inflammation-suppressing effect mediated by ketogenesis catalyzed by HMGCS2 is a protective factor. In addition to being targeted by miRNA, DNA methylation and histone acetylation also participate in the transcription of HMGCS2. Hypermethylation in the CpG island, located in DNA promoter region, and histone deacetylation, was identified in renal cell clear cell carcinoma, resulting in lower β-OHB [43]. PPARγ, an important transcription factor, binds to the HMGCS2 promoter in intestinal epithelial cells [44]. Therefore, HMGCS2 is regulated by multiple epigenetic factors. Also, the activity of HMGCS2 was lower than that of the corresponding normal cell lines in poorly differentiated cancers [45,46], but elevated in higher-grade steroid-independent cancers [47,48].

### 2.3. HMGCL

A significant decrease in HMGCL expression has been observed in lung cancer, prostate cancer, and renal clear cell carcinoma [49,50,51]. HMGCL inhibits the metastasis and invasion of nasopharyngeal carcinoma cells through mesenchymal–epithelial transition (MET) [52]. The modification of H3K27ac in HMGCL induces the transcription of transcription factor FOXM1, thereby mediating glioblastoma proliferation [53]. In addition, disruption of HMGCL significantly reduces pancreatic ductal adenocarcinoma growth in vivo [54]. In contrast, the oncogenic BRAF V600E selectively promotes BRAF V600E-burdened tumor development through the OcT-1-HMGCL-aceto-acetate axis [55]. HMGCL increases H3K9ac levels by producing β-OHB in a dose-dependent manner, therefore leading to ferroptosis [56]. These studies distinctly designate HMGCL as an exciting target for cancer therapy.

### 2.4. BDH

BDH plays a key role in the interconversion of AcAc andβ-OHB during fatty acid metabolism, with two family members, BDH1 and BDH2. BDH1 induces autophagy and subsequent proliferation and migration of lung cancer cells by activating the PARP1-mediated AMPK-mTOR signaling pathway [57]. Metallothionein 2A binds to the histone deacetylase and helicase HDAC2/CHD4 complex, forming a chromatin R loop that inhibits the transcription of BDH1 in hepatocellular carcinoma stem cells [58]. BDH1 expression is significantly downregulated in both adult and pediatric glioblastomas [59]. Interestingly, patients with low BDH1 expression respond better to KD treatment. These findings support the potential use of BDH1 inhibitors to enhance the function of KD in cancer treatment [60,61]. Another member of the BDH family, BDH2, is downregulated in various types of cancer. It controls the ROS levels, thereby restraining the growth of gastric carcinoma [62]. BDH1 and BDH2 have contradictory effects in lung adenocarcinoma. BDH1 promotes the malignant cell phenotype by mediating the H3K9bhb/LRRC31 axis [63], while BDH2 functions as an anti-cancer agent by facilitating cell autophagy and apoptosis [64].

### 2.5. OXCT1

OXCT1, also known as succinyl-CoA transferase (SCOT), exhibits varied expressions in different types of cancer. It is downregulated in various brain tumors [59] and acts as a suppressor in glioma progression through OXCT1-AS1 [65]. CIRC-OXCT1 reduces Smad4 and inhibits Epithelial–Mesenchymal Transition (EMT) in gastric cancer cells [66]. On the contrary, OXCT1 is upregulated in non-small cell lung cancer and prostate cancer [47]. The expression of OXCT1 was higher in cell lines with high mobility than those with low mobility [67]. Recently, OXCT1 was shown to participate in protein modification. It promotes succinylation of Lactamase Beta (LACTB)-K284, thereby blocking its catabolic activity and enhancing mitochondrial respiration, which facilitates hepatocellular carcinoma growth [68]. Moreover, ketogenic fibroblasts support the growth and metastasis of breast cancer cells with overexpression of OXCT1 overexpression [37], suggesting that the tumor microenvironment may become a resource for KBs.

The abnormal expression of key enzymes in KB metabolism pathways in various tumors and their impact on tumor initiation and progression have been described above. As metabolic enzymes, their predominant biological role in cancer biology is significantly influenced by their metabolic products.

## 3. The Potential Function KBs in Tumorigenesis

### 3.1. β-OHB

β-OHB constitutes approximately 80% of KBs and is classified as a class I histone deacetylase inhibitor (HDACi) [69]. It is involved in regulating cell differentiation, proliferation, apoptosis, autophagy, and metastasis via epigenetic reprogramming of the transcriptome [70,71,72] (Figure 2). Apart from influencing histone lysine acetylation (Kac), β-OHB itself can become an epigenetic modifying molecule. A research group from the University of Chicago has reported, for the first time, that a novel post-translational modification-lysine β-hydroxybutyrylation (Kbhb) [73]. Interestingly, the β-OHB-induced Kac and Kbhb sites were not overlapping [74], showing a site-specificity in the modification pattern. The famous tumor suppressor p53 could be Kbhb modified upon the treatment of β-OHB. A higher level of p53-Kbhb leads to a decreased of p53-Kac, thus reducing cell cycle arrest and apoptosis [75]. Whether β-OHB is an anti-cancer or cancer-promoter agent remains a puzzle. It has been shown that β-OHB reprograms energy metabolism, thus inhibiting the proliferation of epithelial cells in colon crypts, as well as colorectal cancer cells [17,76]. Paradoxically, the concentration of β-OHB elevated in the metabolites of colorectal cancer and facilitated tumor growth and distant metastasis [18]. This is because the subjects of these experiments differ. For instance, some studies detected high levels of β-OHB in serum from colorectal cancer patients, while others conducted research on mouse models of KD or colon cancer cell lines derived from mice. The expression of KB metabolism enzymes in these tumors themselves may vary, therefore resulting in an opposite conclusion. In glioma, β-OHB was found to interfere with NOD-like Receptor Family Pyrin Domain Containing 3 (NLRP3) inflammasome activation [77], which further confirms its ability in anti-inflammation. Additionally, β-OHB enhances the T-cell response in viral lung infections [78]. This might promote immune killing effects on tumors as well. However, someone has identified that the physiological level of β-OHB failed to hinder the growth and response to chemo- or radiotherapy in breast cancer [79].

### 3.2. AcAc

AcAc constitutes about 20% of the total KBs in normal individuals. AcAc serves as an optional fuel for glucose, preventing a pseudo-starvation state during lactic acidosis and potentially increasing tissue resistance to acidosis [80]. Increased utilization of AcAc has been observed in the hearts of diabetic animals [81]. Based on this, AcAc was considered a prognostic biomarker for heart failure [82]. Intraventricular injection of AcAc inhibited the expression of inflammatory factors and improved memory in Alzheimer’s mice [83]. Regarding cancer, limited studies related to AcAc have been reported to date. In hepatocellular carcinoma, β-1,3-N-acetylgalactosaminyltransferase II (B3GALNT2) downregulates the expression of AcAc-related metabolic enzymes. The reduction in AcAc enhances the activity of the macrophage migration inhibitory factor (MIF), which ultimately leads to increased macrophage recruitment and promotes tumor growth [19]. However, elevated serum AcAc levels can enhance the growth of human melanoma cells expressing the BRAF V600E kinase mutation in xenograft mice [84].

### 3.3. Acetone

AcAc can undergo spontaneous decarboxylation to form acetone, which accounts for only 2% of KBs. As a volatile organic compound found at high levels in human exhaled breath, acetone could be detected in respiratory metabolomics studies and has emerged as a potential biomarker for the disease. It has been shown to significantly increase acetone levels in patients with liver cirrhosis, liver cancer, and pulmonary hypertension while decreasing in patients with liver metastasis from colorectal cancer [20].

At present, our understanding of AcAc and acetone in tumors is limited, primarily due to the low proportion of KBs. Despite extensive research on β-OHB, it remains unclear that the concentration and dynamic changes in β-OHB in actual solid tumor tissues, let alone its correlation with tumor stage and prognosis. However, the content of KBs circulating in the body can be influenced by diet and measured, providing opportunities to explore the role of dietary regulation in KB production and its implications for tumor occurrence, development, and treatment. Further research in this area could offer valuable insights into how dietary interventions may impact tumor metabolism and therapeutic responses.

## 4. KD as a Novel Adjuvant Approach for Cancer Treatment

### 4.1. Types of KD

The KD has gained popularity among healthcare providers due to its non-invasiveness, low complication rate in normal tissues, and high efficacy in controlling tumors. A moderate-protein, very low-carbohydrate, and high-fat diet mimics metabolism during fasting. There are several variations in the KD (Table 3), each with distinct characteristics and applications:

Classic ketogenic diet (KD): It is a high-fat, low-carbohydrate, and moderate-protein diet originally developed to treat epilepsy. Approximately 90% of total caloric intake comes from fats, primarily long-chain triglycerides (LCTs), with only 8% from protein and 2% from carbohydrates. The primary goal of KD is to induce ketosis, a metabolic state where the body burns fat for fuel instead of carbohydrates, which is particularly beneficial for controlling seizures in pediatric epilepsy patients [21].

Medium-chain triglyceride diet (MCTD): The medium-chain fatty triglycerides (MCTs) are metabolized more efficiently than LCTs, leading to a higher production of KBs per kilocalorie of energy compared to long-chain triglyceride-based diets (LCTD). This flexibility allows for greater dietary variety and larger portion sizes, enhancing patient adherence [22].

Modified Atkins diet (MAD): It is a more palatable and less restrictive alternative to the classic KD, specifically designed to enhance adherence and flexibility in dietary management. Typically, the macronutrient distribution in MAD consists of approximately 65% fat, 30% protein, and 5% carbohydrates. This balance facilitates the maintenance of ketosis with greater dietary variety and larger portion sizes compared to the classic KD. MAD is particularly suitable for children with behavioral issues or when adherence to the classic KD is challenging for parents or physicians [23].

Low-glycemic index treatment (LGIT): In LGIT, carbohydrates with a glycemic index (GI) of less than 50 are permitted, allowing a wide range of foods compared to the classic KD. This includes certain meats, dairy products, select fruits, and whole-grain breads. The key mechanism of LGIT is to stabilize blood glucose levels, which in turn can reduce seizure frequency while still maintaining ketosis through fat consumption. This approach provides greater dietary flexibility while still promoting ketosis [24].

### 4.2. Potential Mechanisms of the KD in Suppressing Cancer

Originally proposed in 1920 for the treatment of intractable epilepsy, KD was later found to interfere with the Warburg effect and inhibit cancer cell growth [85]. Recent studies indicate that exposure to environmental pollutants, such as copper (Cu), can significantly alter cellular metabolism and promote oxidative stress, similar to the metabolic reprogramming observed in cancer cells under KD conditions [86]. KD affects cancer cells through both extracellular and intracellular pathways. Extracellularly, KD lowers blood glucose levels, reducing the production of insulin and insulin-like growth factor 1 (IGF-1), and blocks the mTOR pathway [87]. Intracellularly, tumor cells face challenges in generating ATP through the TCA cycle due to deficiencies in key mitochondrial enzymes required for KB metabolism [25]. This shifts the glycolysis of tumor cells to the mitochondrial aerobic metabolism in healthy cells [88].

Metabolism reprogramming in cancer cells, combined with environmental pollutants and mitochondrial dysfunction, can lead to sustained increases in mitochondrial ROS levels. This further supports the connection between metabolic dysregulation, oxidative stress, and cellular toxicity, which could also be relevant to the efficacy of KD in cancer treatment. Additionally, downstream metabolites of KBs, such as acetyl-CoA and succinyl-CoA, function as signaling molecules that link the external environment to cellular processes, including gene expression regulation [26].

### 4.3. Animal Models of KD in Cancer

Under strict supervision, KD has become a potential dietary approach for cancer patients, which is safe and feasible, making it a prospective adjuvant for multifactorial therapies tailored to individual patients (Table 4). For example, KD showed an inhibition of tumor growth and prolonged survival of lung cancer, derived xenografts models, and hepatocellular carcinoma-bearing mice [89,90]. This effect is likely due to reduced glucose availability and the shift towards KBs as an alternative energy source, which preferentially affects cancer cells that rely heavily on glucose for survival [89,90].A recent meta-analysis of 38 animal studies revealed that the KD significantly prolonged survival time, and reduced tumor weight and volume, with the most notable effects observed at a fat-to-carbohydrate ratio of 4:1 [91]. In [89] a dehydroepiandrosterone (DHEA)-induced polycystic ovary syndrome (PCOS) mouse model, KD significantly increased blood ketone levels, reduced blood glucose and body weight, and improved ovarian function, particularly after three weeks of intervention. Additionally, KD suppressed inflammation and apoptosis in the ovaries, supporting its potential use in metabolic diseases and cancer treatment [92]. In nitrosamine reduction-induced lung cancer, fish oil (FO) has been found to be superior to KD, rich in other fats [93]. Breast cancer patients undergoing radiotherapy who followed the KD showed improved metabolic indicators, reduced insulin resistance, and significantly improved quality of life [94,95]. In pancreatic cancer, KD has been shown to reduce tumor growth and prolong survival in mouse models. This effect is believed to result from the modulation of key signaling pathways such as mTOR and Akt, which are critical for cell proliferation and survival. By shifting the tumor’s metabolic dependence from glucose to KBs, KD may weaken the tumor’s capacity to sustain rapid growth and resistance to therapy [96]. These findings demonstrate that the KD’s effects on tumor cell metabolic flexibility and proliferation vary by cancer type, but generally operate by disrupting tumor cells metabolic flexibility, limiting glucose availability, and inducing oxidative stress and apoptosis, collectively supporting KD’s potential as an adjunctive therapy in cancer treatment.

Apart from its independent utilization, the KD could be integrated with other treatments. KD has demonstrated valuable potential in improving the effectiveness of immunotherapy for various cancers [109]. The administration of KD has been found to bolster therapeutic responses to anti-programmed cell death l (PD-1) treatment alone or when combined with anti-Cytotoxic T-lymphocyte-associated protein 4 (CTLA-4) in aggressive tumor models by reinstating T cell-mediated immunosurveillance [5]. The activation of the AMPK signaling pathway triggered by KD leads to programmed cell death-ligand 1 (PD-L1) phosphorylation and degradation. Furthermore, KD disrupts the function of the polycomb repressive complex 2 (PRC2) transcriptional complex, consequently boosting the expression of interferon and antigen-presenting genes [110]. Moreover, the composition of the gut microbiota has been demonstrated to be significantly altered by KD, favoring a shift from tolerogenic bacteria (such as Lactobacillus) to immunogenic bacteria (such as Myxobacteria) [111]. These findings indicate that KD remarkably improves the function of immune cells, thereby facilitating the immune checkpoint therapy effect in cancer patients. When evaluating the impact of nutritional interventions on cancer, it is crucial to assess not only metabolic and inflammation-related index but also immunological parameters [112].

### 4.4. Preclinical and Clinical Studies of KD in Cancer

Recent animal studies have demonstrated that the KD holds promise as a cancer therapy [113]. Additionally, several preclinical studies suggest that KD interventions possess potent anti-cancer effects; however, the relationship between KD and cancer in clinical trials remains unclear [114]. Here, we focus on preclinical and clinical trials conducted between 2020 and 2024 (Table 4). In preclinical studies, KD has been shown to enhance the response to radiation therapy in pancreatic xenograft models [102]. Furthermore, KD, with radiation, exhibited a slight enhancement in tumor growth and prolonged survival in head and neck cancer models [106]. However, the responses varied depending on the diet formulation [106]. In clinical studies, most research has concentrated on highly aggressive cancer types with poor prognosis, such as high-grade gliomas, pancreatic cancer, triple-negative breast cancer, and advanced or metastatic cancers [102,103,106]. The majority of the presented data indicate that KD was a supplement to chemotherapy or radiation, while in some clinical studies, KD was employed as the sole treatment [107].

These clinical studies primarily focused on metabolic changes, safety, tolerability, and quality of life of KD. KD significantly increased β-OHB levels, and reduced glucose and insulin levels [103,115]. However, some clinical studies observed no significant changes in glucose [94]. Moreover, several studies reported potential improvements in the overall health status and quality of life of patients [116].

One clinical trial suggests that KD may not be effective as a standalone therapy; rather, it should be used in conjunction with other therapeutic approaches, such as chemotherapy and radiotherapy [117]. In one randomized controlled trial, KD was incorporated as a complementary treatment alongside chemotherapy. In patients with locally advanced disease, a decrease in the TNF-α and insulin was accompanied by a decrease in tumor size and staging [103]. Combining KD with cytotoxic chemotherapy has exhibited the ability to impede pancreatic cancer growth [96], and enhance overall survival in breast cancer patients without notable side effects [29]. KD has shown potential in improving the efficacy of chemotherapy and radiotherapy in brain tumors, rectal cancer, and liver cancer [118,119,120,121].

### 4.5. Limitations and Risks of Clinical Studies on the KD in Cancer

The KD has shown potential in inhibiting cancer cells, animal models, and preclinical studies, but its clinical application in cancer therapy is hindered by several important limitations and potential adverse reactions [27]. Most clinical studies on KD are characterized by small patient populations, a lack of long-term follow-up, and significant variability in study designs. Differences in cancer types, patient characteristics (e.g., age, sex, and overall health), and KD protocols across trials complicate the interpretation of results. Many studies rely on self-reported data regarding dietary composition without concurrent measurements of blood or urine ketone levels, raising doubts about whether ketosis is achieved, and undermining the ability to accurately assess the effects of KD. These make it difficult to assess the sustained efficacy and safety of KD over time [122].

In addition, KD is associated with various potential side effects, including gastrointestinal issues, elevated lipid levels, kidney stones, and nutrient deficiencies [28,29]. The long-term health consequences of KD remain unclear, as few studies track patient health after the diet is discontinued. The diet may also have varying effects on different individuals, depending on factors such as pre-existing health conditions, genetic factors, and tumor types. Moreover, KD’s potential to enhance anti-tumor immunity while potentially impairing immune function complicates its use as a universal therapeutic approach [30,31]. The strict dietary requirements and reliance on high-fat foods may also reduce patient adherence [31,123], particularly in cultures with carbohydrate-rich diets.

Despite promising preclinical evidence, the human trials conducted thus far have yielded inconclusive results. This underscores the necessity for large-scale, well-designed, randomized controlled trials to assess the efficacy, safety, and long-term effects of KD as an adjunctive cancer therapy. Research should focus on understanding the molecular mechanisms behind KD’s effects, tailoring the diet to specific tumor types, and developing alternative KD variants to improve patient tolerance.

### 4.6. Next Step of KD in Clinical Application

To thoroughly understand the role of KD in tumor treatment and evaluate its clinical application, several key steps should be prioritized in future research and clinical trials. Firstly, molecular studies should elucidate the mechanisms by which KD affects tumor cells, including its impact on metabolic pathways, immune responses, and cancer-related signaling pathways. Secondly, randomized controlled trials with diverse study designs are necessary to provide robust evidence on KD’s efficacy and safety across different types of tumors. Additionally, high-quality clinical data are essential to support KD’s effectiveness and safety profiles in diverse patient populations. Further exploration of dietary modifications to minimize side effects and enhance patient tolerance is crucial. Personalizing KD therapy involves identifying specific symptoms, conditions, and tumor types that may benefit from KD, understanding contraindications, and developing well-tolerated variants of KD. Lastly, conducting clinical trials across various tumor types will comprehensively assess KD’s efficacy and safety, optimizing its clinical application in oncology.

## 5. Conclusions and Future Prospects

There is growing interest in the metabolism of KBs in tumors. Identifying dysregulation of key genes involved in KB metabolism can provide valuable insights into the diagnosis and prognosis of malignant tumors. The primary component of KBs, β-OHB, has shown significant tumor-suppressive effects in many cancers and influences the transcriptional profile of tumor cells through epigenetic mechanisms. Significant progress has been made in understanding the mechanisms of KB metabolism and the potential therapeutic applications of KD in tumor therapy, positioning it as a promising strategy for targeting aberrant tumor metabolism. However, numerous questions remain unanswered. These include understanding how KB metabolism changes during tumor development, elucidating the mechanisms underlying the dysregulation of key KB metabolic enzymes, exploring the intricate interplay between KB metabolism and epigenetic modifications, and addressing the inconsistent effects of KD as an adjuvant therapy to enhance tumor treatment. Further research is essential to clarify the clinical indications for KD application in cancer therapy.

## Figures and Tables

**Figure 1 biomedicines-13-00210-f001:**
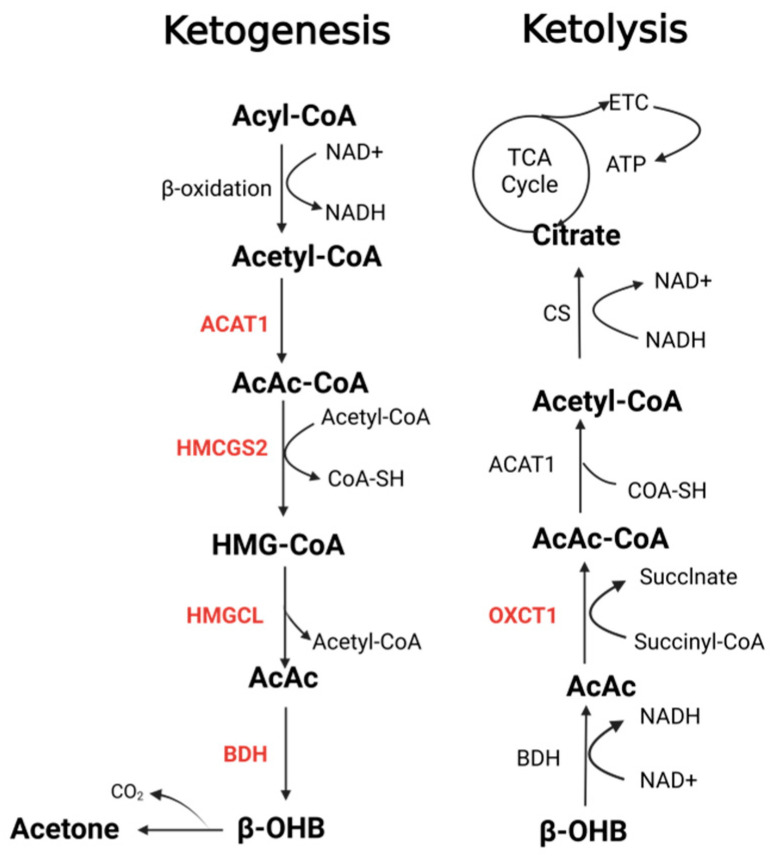
Metabolic pathway of KBs.

**Figure 2 biomedicines-13-00210-f002:**
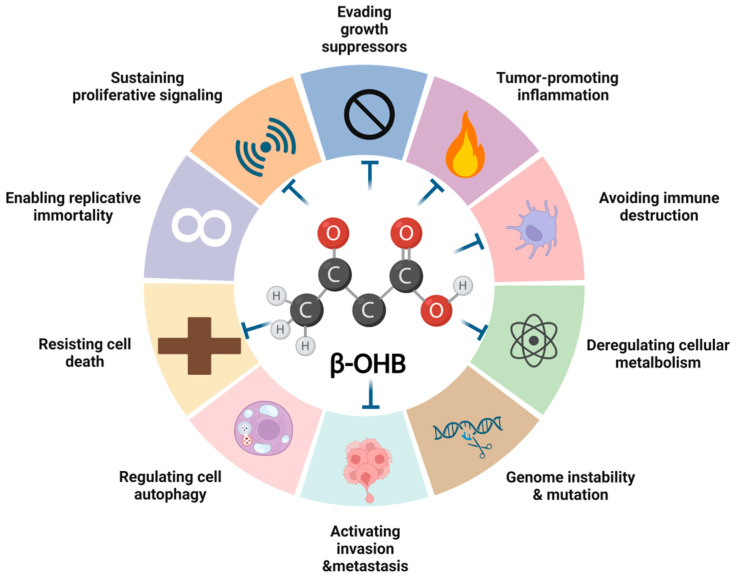
The effects of β-OHB on hallmarks of cancer.

**Table 1 biomedicines-13-00210-t001:** KD food list.

Food Category	Recommended Foods	Foods to Avoid
Fats	Olive oil, coconut oil, butter, avocado oil, nuts	Vegetable oils (e.g., corn oil and peanut oil)
Protein	beef, pork, chicken, and fatty fish (e.g., salmon)	Lean meats (e.g., chicken breast and lean beef)
Vegetables	Leafy greens, cauliflower, cucumber, and mushrooms	Starchy vegetables (e.g., potatoes and corn)
Fruits	Berries (strawberries, blueberries), lemon, and avocado	High-sugar fruits (e.g., bananas and apples)
Dairy	Full-fat dairy (milk, cheese, butter, and yogurt)	Low-fat dairy (e.g., skim milk and low-fat yogurt)
Nuts/Seeds	Almonds, walnuts, pumpkin seeds, and chia seeds	Sugary nuts (e.g., certain mixed nuts)
Beverages	Water, unsweetened coffee, and tea	Sugary drinks (e.g., sodas and fruit juice)
Condiments	Salt, spices, vinegar, and lemon juice	Sugary sauces (e.g., ketchup and salad dressing)

**Table 2 biomedicines-13-00210-t002:** Summary of KB metabolism and KD in cancer: a comprehensive overview.

Section	Content	Clinical Implications	References
Dysregulation of KB Metabolism Enzymes	Dysregulation of key enzymes (ACAT1, HMGCS2, HMGCL, BDH, and OXCT1) in tumor cells affects KBs synthesis, leading to altered metabolic processes in tumors.	Dysregulated enzymes can serve as biomarkers or therapeutic targets for cancer diagnosis and treatment. Inhibition of specific enzymes may slow tumor growth.	[15,16,17]
Potential Role of KBs in Tumorigenesis	β-OHB, AcAc, and Acetone play roles in cell differentiation, inflammation, energy metabolism, and immune response. The exact role of β-OHB in cancer is complex.	KBs, especially β-OHB, have both tumor-suppressive and tumor-promoting effects, depending on concentration, tumor type, and metabolism. Requires further study.	[18,19,20]
KD Overview	KD mimics fasting metaboh high fat, low carbs. Variants (MCTD, MAD, and LGIT) provide flexibility. A 4:1 fat-to-carb+protein ratio promotes ketogenesis.	KD is an emerging therapy for cancer, targeting tumor metabolism and promoting alternative energy utilization. Affects tumor growth, immune response, and therapy.	[21,22,23,24]
KD and Tumor Metabolism	KD reduces glucose and insulin, shifts tumors to ketone metabolism, inhibits mTOR and TCA cycle, weakening tumor growth.	KD inhibits tumor growth and enhances the effectiveness of chemotherapy and radiotherapy by limiting glucose and promoting oxidative stress in tumor cells.	[25,26]
Combination with Immunotherapy and Chemo	KD boosts immune responses, enhances checkpoint inhibitors (PD-1 and CTLA-4), and supports chemotherapy by reprogramming immune and metabolic pathways.	KD enhances immune therapy responses and synergizes with chemotherapy to improve survival and reduce tumor size in various cancers (e.g., breast, pancreatic, and lung).	[5]
Side Effects and Challenges	GI discomfort, nutrient deficiencies, fatigue, and the potential for increased lipid levels or kidney stones. Adherence can be difficult, especially long-term.	Need for personalized care, addressing side effects, and adjusting KD protocols for individual tolerance and tumor type.	[27,28,29]
Future Research Directions	Explore KD’s detailed effects on tumor metabolism, immune response, enzyme regulation, and its long-term impact on cancer treatment.	More high-quality, large-scale studies needed to understand optimal KD use in cancer therapy, and to address its limitations and potential adverse effects.	[30,31]

**Table 3 biomedicines-13-00210-t003:** Summary of Different Types of KD.

KD Type	Fat: Carbohydrate + Protein Ratio	Main Components	Target Population	Key Features	References
Classic Ketogenic Diet (KD)	4:1	LCTs	Children with epilepsy, and refractory epilepsy patients	Strict carbohydrate restriction and effective for epilepsy control	[21]
Medium-Chain Triglyceride Diet (MCTD)	3:1	Medium-chain fatty acids (e.g., caprylic, capric acid)	Epilepsy and metabolic disorders	Higher ketone production per calorie and more efficient energy use	[22]
Modified Atkins Diet (MAD)	1:1 or 2:1	Animal proteins, low-carb vegetables, fats	Children, adolescents, and adults	More flexible, palatable, and suitable for long-term use	[23]
Low-Glycemic Index Treatment (LGIT)	1.5:1	Low glycemic index foods (e.g., low GI grains, selected fruits)	Epilepsy and diabetes patients	Greater dietary flexibility and still promotes ketosis	[24]

**Table 4 biomedicines-13-00210-t004:** The impact of KD in cancer.

Type	Cancer	Clinical Trial No./Animal Models	Conbined with Tumor Therapy	Study Duration	Metabolic Levels of the KD Group (Upregulated ↑; Downregulated ↓)	Major Outcome of the KD Groups	References
Animal models	Breast cancer	BALB/c mice		35 days		Inhibited the proliferation of 4T1 tumor cells in vivo and temporarily slowed the growth of 4T1 primary tumors.	[97]
	Polycystic ovary syndrome	C57BL/6 female mice		1 weeks and 3 weeks	bloodβ-OHB ↑, blood glucose ↓, body weight ↓	Inflammation and apoptosis in the ovaries of mice treated with DHEA+KD are suppressed.	[92]
	Pancreatic cancer	Six-month KC mice		6 months		KD modulated insulin signaling and hepatic lipid metabolism, highlighting its beneficial effects on health span and liver function compared to HFD.	[98]
	Pancreatic ductal adenocarcinoma	12-week C57BL/6J mice		2 weeks		KD improved the tibialis anterior muscle fiber diameter, circulating KBs, and Hmgcs2 expression levels in PDAC mice.	[99]
	glioblastoma	U87 glioblastoma mouse models	Bevacizumab	80 days		The combined therapy of KD and Bev shows a decrease in tumor growth rate and an increase in mouse survival time.	[100]
	Colon cancer	BALB/c mice		15 days	blood β-OHB ↑, blood glucose ↓	KD can prevent the progression of colon tumors by inducing oxidative stress within the tumor, inhibiting the expression of MMP-9, and enhancing the polarization of M2 to M1 type TAMs.	[101]
Preclinical study	Lung cancer and pancreatic cancer	Phase 1 Trial (NCT01419587)	Radiation	Lung cancer: 5 weeks, Pancreatic Cancer: 6 weeks		KD enhances radiation therapy response in a pancreas xenograft model. KD increase immuno-reactive 4HNE-modified proteins in pancreas xenograft tumor tissue.	[102]
Clinical study	Breast cancer	Randomized controlled trial (IRCT20171105037259N2)	Chemotherapy	12 weeks	TNF-α ↓, IL-10 ↑, serum insulin ↓	A reduction in tumor size in the KD. Stage decreased significantly in patients with locally advanced disease in the KD.	[103]
	Breast cancer	Randomized controlled trial (IRCT20171105037259N2)	Chemotherapy	12 weeks	serum lactate ↓; ALP ↓		[104]
	Glioma	Phase I clinical trial (NCT02149459)	Radiation	2 weeks	β-OHB ↑, ketones in urine ↑, glucose ↓		[105]
	Head and neck cancer	Phase 1 trial (NCT01975766)	Chemotherapy	75 days and 120 days		Mice receiving radiotherapy and KD show slight improvement in tumor growth rate and survival rate.	[106]
	Ovarian and Endometrial cancer	Randomized Controlled trial (NCT03171506)		12 weeks	β-OHB ↑		[107]
	Rectal cancer	Non-randomized, controlled Trial (NCT02516501)	Radiation	55 days	β-OHB ↑; body weight ↓, fat mass ↓		[108]

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
