# Peer review of "Comprehensive Overview of Ketone Bodies in Cancer Metabolism: Mechanisms and Application"

_biomedicines, 2025, doi:10.3390/biomedicines13010210_

Round 1

Reviewer 1 Report

Comments and Suggestions for Authors

The authors present an important and relevant study. This work is timely and of significant interest not only to the academic community but also to other audiences. It is well-written, current, and presented in an accessible manner. However, there are some aspects that, in my opinion, could be improved.

Please define the objective of the study more clearly. Additionally, consider revising the title to better align with this objective. While the current title is descriptive, it could be more direct and reflective of the study's main focus.

Methodology: Given the contemporary nature of the topic and the limited availability of an extensive bibliography, it would be beneficial to include details on how the research was conducted. For example: How was the literature search performed? Which studies were included? What keywords and search engines/databases were used? Although this is not a meta-analysis, providing such context would help the reader better understand the scope of the existing literature and its timeline. Specifically, how many studies are available, and from which year did this field gain more attention?

For statements like, “They could potentially serve as diagnostic and prognostic biomarkers of cancers, as well as therapeutic targets,” consider adding specific references. What types of cancer are being referred to in this context? Providing citations will strengthen the argument and give readers a clearer understanding.

The article could benefit from some restructuring. As I mentioned earlier, it is essential to clearly define the paper's objective. This will help organize the content more logically. The objetive is more the role of ketone bodies in tumorigenesis, their potential functions, or the future therapeutic application of the ketogenic diet?

Including a summary table could greatly enhance the clarity of the article. This table could summarize the main findings from the literature, categorizing them under key themes such as ketone body functions, impacts on tumors, and potential therapeutic applications.

Consider adding a brief discussion of the current challenges and limitations associated with applying the ketogenic diet as a cancer therapy. 

By clarifying these aspects, the article will become more comprehensible and impactful.

Reviewer 2 Report

Comments and Suggestions for Authors

This interesting review focuses on describing the potential benefits of the ketogenic diet in cancer.

Comments:

1.  It is recommended to describe in detail the differences in the metabolism of tumor cells and normal cells, and through what mechanisms the ketogenic diet affects the tumor, the metastases, and the cells of the tumor microenvironment

2.   It is recommended to describe in detail what types of ketogenic diets there are and add a table.

3.   It is recommended to describe in more detail how the ketogenic diet affects the proliferation of different tumor types

4. it is recommended that Table 1 be expanded with results from preclinical and clinical studies, e.g. from ClinicalsTrials and discussed in the text. From the current Table 1, the advantages or disadvantages of the ketogenic diet are not clear or the description is very generalized.

5.  It is recommended to describe the clinical limitations of the ketogenic diet in more detail

Reviewer 3 Report

Comments and Suggestions for Authors

The review provides, a compelling overview of the role of ketone bodies in cancer metabolism, emphasizing their multifaceted contributions to tumor development and treatment sensitivity. The exploration of KBs as signaling mediators and their impact on post-translational modifications, inflammation, and oxidative stress is particularly intriguing. The integration of ketogenic diet (KD) strategies into adjunct cancer therapy adds translational value to the discussion. However, the manuscript would benefit from additional specificity and contextual clarity in several areas to strengthen its contribution to the field. The review covered recent literature with good graphical/illustrative representation of published work.

The comments to improve the review are as follows:

1.      Abstract: Add brief sentence of mechanism how ketogenic diet help as adjunct cancer therapy.

2.      Introduction:  A paragraph of sentence of metabolic cause of cancer like heavy metal, https://doi.org/10.1016/j.ecoenv.2024.117078, https://doi.org/10.1016/j.biopha.2022.113993 etc

3.      Keto diet food table should be there in introduction.

4.      Include clinical study data in tabular form. There is only one tabular data at present.

5.      Add few research study data in animal with permission to improve the understanding of topic.

6.      Add limitation of ketogenic diet/side effect/control if required as a diet.

7.      Overall content should be increase as at present the length is like mini review.

Round 2

Reviewer 1 Report

Comments and Suggestions for Authors

 The authors have answered the questions satisfactorily. I propose that the article be accepted. 

Author Response

Dear Reviewer,

I would like to express my sincere gratitude for your thoughtful review and acceptance of my manuscript. It is truly an honor to receive your approval. I greatly appreciate the time and effort you dedicated to reviewing my work, and your valuable feedback has significantly contributed to improving the quality of the paper. I will continue to strive for further improvements in my future research.

Thank you once again for your hard work and support!

Sincerely yours,

Weililn Zhao

Reviewer 2 Report

Comments and Suggestions for Authors

The authors answered my questions and made corrections to the manuscript, which improved the quality of the manuscript. It is recommended to add a column with references to the source of literature in Tables 2 and 3. All abbreviations should be written in full when first mentioned.

Reviewer 3 Report

Comments and Suggestions for Authors

Revision is acceptable 

Author Response

(The authors gave the same response as above.)
